# Experience of Cannabis Use from Adolescence to Adulthood in France: An Interpretative Phenomenological Analysis

**DOI:** 10.3390/ijerph20054462

**Published:** 2023-03-02

**Authors:** Selma Faten Rezag Bara, Murielle Mary-Krause, Solène Wallez, Jean-Sébastien Cadwallader

**Affiliations:** 1Sorbonne University, INSERM, Institut Pierre Louis d’Épidémiologie et de Santé Publique (IPLESP), F75012 Paris, France; 2Department of General Practice, Sorbonne University, F75012 Paris, France

**Keywords:** cannabis, therapeutic, consumption, adults, interpretative phenomenological analysis, qualitative methods

## Abstract

Levels of cannabis use are high during adolescence, but the proportion of cannabis users among adults is also progressing, often for medical reasons. This study describes the reasons and motivations for using medical cannabis among adults over 30 years old in France. This qualitative study was performed using an interpretative phenomenological analysis. People with a history of cannabis use or current cannabis users were recruited from the TEMPO cohort. Homogeneous purposive sampling was applied among those using medical cannabis. Twelve participants, among thirty-six who reported using cannabis for medical reasons, were selected and interviewed. Five superordinate themes were identified in the analysis: 1—soothing a traumatic experience through cannabis use; 2—an ambivalent relationship with the user and cannabis and with the user and close relatives; 3—cannabis, a known soft drug comparable to alcohol or tobacco, leading to an illogical demonization; 4—recreational use in the context of experimentation; and 5—a paradoxical desire for exemplary parenting. In this first recent study to describe the reasons and views adults have in order to continue using cannabis after 30 years of age, we identified ways to explain this consumption. The internal appeasement provoked by cannabis stems from a struggle to appease a violent external situation.

## 1. Introduction

As in other countries, cannabis is the most experienced and used illicit drug among the French population [1]. In 2021, 47.3% of 18–64-year-olds had already experimented with cannabis (32.9% in 2010), with 10.6% continuing to use it in the following year (8.0% in 2010), in a ratio of 1 woman for every 2 men [2]. Problematic use can be deduced from the overall number of patients admitted to treatment for the first time for cannabis-related problems, which increased by 76% between 2006 and 2017 in Europe [3]. Nevertheless, while France has one of the most repressive illegal drug legislations in Europe, this also the European countries with the highest levels of cannabis use (44.8% of lifetime users aged 15–64 vs. 37.5% in Spain or 22.6% in Belgium, even 6.1% in Hungary; 21.8% of past-year users aged 15–34 vs. 19.1% in Spain or 13.6% in Belgium, even 3.4% in Hungary) [1]. Different legislation in European countries could lead to different cannabis use [4]. A recent study showed that older adults experienced recreational cannabis for medicinal purposes following legalization [5]. However, there was no evidence that reasons to use cannabis for medicinal purposes changed according to cannabis legislation.

If the levels of cannabis use are high during adolescence, the proportion of cannabis users among adults is progressing even more, reflecting both the aging of the generations that experimented with this product during a period of high popularity, alongside a decreasing rate of cannabis initiation among younger generations [6]. In France, daily use is increasing among older generations: from 1.4% in 2014 to 2.0% in 2017 for 35–44-year-olds and from 0.6% to 1.2% for 45–54-year-olds [7]. Over the past 30 years, observations of cannabis use in the adult population have revealed a trend: the average age of experimental users and users during the year was increasingly linked to people in their thirties and forties. This trend clearly suggests that some of the first generations of users did not give up their cannabis use as they became older. This increase in use throughout the French population therefore significantly alters the demographic distribution of cannabis consumers [8].

Otherwise, the high risk of “problematic” use peaks at 28% for users aged between 26 and 44 years old [7]. However, this level varies depending on age: between 45 and 64, more than one out of five users also have an increased risk of problematic use [7]. Thus, among people seeking help from healthcare centers, the percentage of cannabis users over the age of 40 increased from 5.4% in 2007 to 9.6% in 2017 [9,10].

The reasons for cannabis use among adults are unclear. In particular, during adulthood, cannabis may be used by people with emotional and/or psychological difficulties, in addition to, or instead of, psychotropic drugs. Indeed, 52% of people aged 50 and over use cannabis for medical purposes and 18% recreationally, compared to 18% and 50%, respectively, among 18–29 year-olds [11]. Other studies show that the main reason for using medical cannabis is pain (52.5% of people) with a higher percentage among those aged 45 and over (60.9%) compared to those aged 25–44 (45.4%) [12]. Other medical reasons for using cannabis include anxiety, nervousness and depression for 18.8% of the subjects and insomnia for 18.3% [10].

A recent qualitative study conducted in the US state of Colorado aimed to identify reasons for medical and recreational cannabis use and perceptions of cannabis among people over 60. The study showed that the primary reason of using cannabis was pain management [13]. Some used it as an alternative of other treatments, such as opioids [14].

Between 2014 and 2017, the French Observatory of Drugs and Addictive Tendencies (“Observatoire français des drogues et des tendances addictives”, OFDT) conducted a qualitative study called ARAMIS to better understand the motivations of young people to experiment and consume psychoactive substances while retracing their consumption trajectories [15]. Experimentation with cannabis, unlike cigarettes, gave rise to positive impressions and, very often, benefited from the image of being less addictive and less “dangerous” [15]. Nevertheless, the trajectories of cannabis consumption from adolescence to adulthood remain poorly understood and the perceptions and views of this drug among adults have not yet been documented in France. Recreational cannabis has often been perceived more negatively than medical cannabis, with views of cannabis being influenced by the way the substance is consumed [13].

Our aim was to determine the reasons and motivations for using medical cannabis among adults over 30 years of age in France.

## 2. Materials and Methods

In part of a mixed-methods research study, we carried out a qualitative investigation using an interpretative phenomenological analysis (IPA). This type of analysis was used to offer insight into individual experience and the participants’ views in order to identify a phenomenon common to all [16]. We used the COREQ-32 criteria to ensure the validity of our study [17].

### 2.1. Sampling

The study was based on data from the TEMPO cohort, a cohort of young adults aged 25 to 47 (40 on average in 2020), followed longitudinally since 1991 with successive data collection interviews in 1999, 2009, 2011, 2015, 2018 and 2020–2021, during the COVID-19 pandemic [18]. In 2021, the TEMPO cohort included 659 participants. Among them, 58% used cannabis at least once in their lifetime. Detailed reasons were available for 91% of them.

Homogeneous purposive sampling was applied among participants who reported using cannabis for medical reasons. The definition of the medical use of cannabis in the qualitative study was based on participants’ answers in the quantitative study: self-medication to manage stress, anxiety, headaches or migraines, chronic pain, depression, muscle spasms, nausea, loss of appetite, muscle stiffness, epileptic seizures, tremors or to prevent vomiting. We sought to ensure that participants’ characteristics were as diverse as possible.

### 2.2. Sample

One third of the 36 participants who used cannabis for medical reasons were interviewed. Between January and May 2022, we carried out 12 in-depth, individual interviews in French, each lasting around two hours.

During these interviews, we collected information concerning their social-emotional life, professional life, housing and access to healthcare [19].

### 2.3. Data Collection

After written consent was given by the participants, comprehensive, in-depth, individual video interviews were recorded, anonymized and transcribed. These interviews were conducted freely following the annotations of an interview guide drafted by researchers with clinical and cannabis-related expertise (see Appendix B). It included questions about participants’ cannabis consumption, e.g., quantity and frequency, motivations for consumption and questions about how their consumption impacts their daily life.

All the researchers kept a logbook throughout data collection to record their feelings and preconceptions (Appendix A). The main preconception was: adults consumed cannabis to treat physical illnesses if/when they had tried all the treatments available. Specifically, we hypothesized that people who use cannabis over the age of 30 do so mainly for medical reasons.

### 2.4. Data Analysis

Three phases of analysis were carried out as part of the IPA method [16]. We first analyzed the interviews one by one, coding the verbatim line by line. We assigned codes to groups of words, sentences or paragraphs. From there, 1500 characteristics emerged. Then, we created specific categories in each interview and we finalized 13 specific categories. Once each transcript had been analyzed, a table of superordinate themes was constructed. We obtained 5 superordinate themes. Triangulation between the researchers was carried out at each step of the analysis and after each interview in dedicated sessions. The researchers discussed these codes and reached a consensus. Data sufficiency was sought and obtained with conclusive categories identified from the coded data and new categories emerging from new interviews were similar to the previous ones.

### 2.5. Ethics

The Sorbonne University Ethics Committee approved the qualitative study (n° CER-2021-069). All participants gave their written consent.

## 3. Results

Twelve individuals participated in the present study. Participants had a median age of 41.1 years. Seven were women and three had chronic diseases. Participant characteristics are presented in Table 1.

We identified 13 categories from which 5 superordinate themes emerged (Table 2): soothing a traumatic experience through cannabis use; an ambivalent relationship with the user and cannabis and with the user and close relatives; cannabis, a known soft drug comparable to alcohol or tobacco, leading to an illogical demonization; recreational use in the context of experimentation; a paradoxical desire for exemplary parenting.

### 3.1. Soothing a Traumatic Experience through Cannabis Use

The people interviewed felt uncomfortable and unhappy during their adolescence. They felt bad about themselves and used cannabis to feel better.

For example, for participant 9 (P9), *“There was an awkwardness. Adolescence was a difficult time for me, with a lot of emotional difficulties. And that allowed me to.... To bring down the level, and then put me in a certain state of stability.”*

Many of the participants were unable to cope with violent, traumatic external events in their lives. They smoked joints to escape their victim status, a status they were aware of. This traumatic event could be their parents’ divorce, like for P1, who said: *“One day someone suggests it and then you say yes. You’re not well. You’re in that thing where your parents are getting divorced, at 13, it’s the end of the world.”* This pattern was also seen to stem from other events, such as rape. After that event, P2 experienced a series of relationship breakdowns which she did not know how to handle. Speaking about her rape, she said: *“A big party in the countryside, my parents weren’t there, so [...] I got quite drunk [...]. Two of my father’s friends [...] took me home in the car and then afterwards... they took advantage of me. So, rape or not, it’s complicated. I know I said no [...] but at the same time I was drunk [...] so clearly with hindsight for me, it’s rape.”*

Other participants experienced professional or personal harassment, as well as physical or emotional abuse. This was, for example, criticism about their weight from their parents (P12), or an abusive partner, as for P2: *“I continued on with someone even worse, who really abused me, not physically but [...] morally. I had an abortion.”*

P8 mentioned the loss of a child: *“That’s personal but, I’ve lost a child (cries) so it’s never easy... oh shit... there you go (silence)... And so there are times when it... It just feels good to... To be able to think about something else to be able to... To be more zen... To look at it more relaxed. […] I had a little girl who died at birth. Died* in utero *just before birth (sobs)... It’s an experience I wouldn’t wish on anyone (I don’t wish for anyone) (sobs). I think that the death of a child is not... It’s not in the order of things, we’ll say it’s hard.”*

Participants were affected by these events and continued to suffer from them. During their interviews, P2, P4 (who had both experienced breakups and depression) and P8 cried.

Thus, cannabis was used by many of the participants to treat depression. For them, life was and is a series of continuous struggles. Cannabis relieved them. Smoking cannabis via joints was carried out for self-medication and was even considered *“better”* than the so-called classical treatments. They tried other treatments and did not want to use them, instead identifying the benefits of cannabis use through their personal experience. Antidepressant treatment was considered to be more addictive.

P8: “*But on the other hand, I prefer that to taking antidepressants [...] Clearly. My therapist told me, frankly, in your case... It’s not worth going into depression... And it’s not worth taking antidepressants either. So, if you have something (cannabis) that makes you feel good at the time, it allows you to come down, breathe and relativize...*”

### 3.2. An Ambivalent Relationship with the User and Cannabis and with the User and Close Relatives

When discussing cannabis use, participants explained they knew and understood the risks, but at the same time regretted their consumption and were ashamed of it. Moreover, they may have felt judged for their use: P1 said that she did not appreciate the way others looked at her: *“Because of the people. In fact, I’m the one who’s panicking [...] I say to myself, ‘Oh, there are people around you. My God, there are people around, they’ll see me, they’ll understand.”*

In a similar way, they devalued themselves but also affirmed their strength and their ability to fight their addiction. An addiction that was difficult to fight according to P6: “*In Saint Denis, it’s hard to get out of it (cannabis)*.”

Furthermore, the interviewees both liked and hated cannabis at the same time. They liked the taste and smell but they *“didn’t advocate cannabis”* (P1, P2, P8). They did not want to be cannabis users, for example, P2 said: *“I’m not in favor of cannabis. The consumption I had I don’t want it”*. Nevertheless, they had simultaneously negative and positive perceptions of cannabis, as explained by P8: *“Because I really like the taste of weed, I love all that. I’m a gourmet.”*

Cannabis was also a way to socialize. It allowed for participants to belong to a group. It was the thread that kept them attached to each other, as expressed by: *“Cannabis is a golden thread”*.

It was both a solution to an unease but also a constraint. They felt appeased but it was in fact a *“false”* appeasement.

P6: *“Cannabis as a smokescreen”*

This ambivalence was found in participants’ relationships with their loved ones: they felt supported and supported others, but also highlighted complicated relationships. For example, for P6, his wife is his *“salvation”*. She was the one that helped him stop a consumption that was killing him. However, P1’s daughter was critical of her mother: *“Well, afterwards, it’s silly little thoughts of teenage girls that you... but it’s just to sting... in fact… she’s trying to hurt (laughs)”*.

Loneliness, unspoken words and misunderstandings were recurrent feelings when speaking about their families.

P5: *“In... our family there were, as in any family... things left unsaid”*

This ambivalence was also found in their relationship with doctors. They could trust their doctor but might also be suspicious of him or her, for example, P1 said: *“Of course, he’s a doctor... You know what I mean? He’s not going to tell me to smoke”.*

### 3.3. Cannabis, a Known Soft Drug Comparable to Alcohol or Tobacco, Leading to an Illogical Demonization

All 12 participants displayed knowledge surrounding cannabis. They discussed the forms of consumption, dosage and ways to obtain the drug. They knew the difference between cannabidiol (CBD) and tetrahydrocannabinol (THC).

P8: *“I think that if there was for example CBD in vapes or things like that, like they do in other countries, the risk is almost zero.”*

Some held an activist attitude. They insisted on the awareness they had about the importance of providing education about cannabis.

P8: *“I think that legalizing and raising awareness much earlier, you know, in a... in a collective intelligence way. Not in a judgmental way.”*

Cannabis was seen as a *“natural”* soft drug, unlike other so-called *“chemical”* drugs. One participant even spoke of an *“organic”* product.

P11: *“I mean is it (cannabis) organic... Mainly (laughs) yeah is it organic or has there been chemical input?”*

P12: *“Cannabis is also natural.”*

P12: *“A drug or coke, you are necessarily addicted to it much faster than a natural product (cannabis) in quotation marks.”*

Cannabis was also distinguished from tobacco or alcohol. It was even considered sometimes less addictive, and participants preferred it to the negative effects of alcohol or tobacco. This comparison with drinking alcohol and smoking tobacco was omnipresent. They did not understand why tobacco or alcohol were legal when cannabis was not.

P9: *“Alcohol can cause delirium tremens, you can have very, very violent things actually when you stop drinking and when you have a regular consumption/use of cannabis, it doesn’t do that at all.”*

According to them, cannabis was demonized. They did not understand why cannabis was judged so harshly. They considered this demonization illogical, a demonization advocated by their parents, who grew up in the 60s, a period of high drug diffusion.

P6: *“Cannabis was something that didn’t speak to them very much, even though they were part of the sixties.”*

### 3.4. Recreational Use in the Context of Experimentation

Most of the participants were seeking fun and enjoyment, a feeling of well-being and pleasure. They enjoyed their experience of consumption. For them, cannabis was a symbol of transgression.

P9: *“Initially, it’s a... an experience of discovery, a little transgressive research where we are in something where we experiment, let’s say.”*

The interviewees had a common desire to experiment with different drugs, but preferred cannabis.

P5: *“I dipped into circles that were a bit … one thing leading to another with other narcotics that I didn’t really get hooked on […] ‘ecstasy, methamphetamine, I tried a little MDMA. But that was purely festive … In particular situations … related to either rave parties or private student parties.”*

While discussing their use, participants distinguished the difference between using cannabis when they were young vs. when they were older. Most of them said that *“it was different”* now that they were older. They no longer bought it to use recreationally but used it to manage conditions and symptoms, for both physical and mental illness.

Some of the participants, mainly the men, linked their consumption with violence. Their relationship with cannabis was tainted with violence, such as previously dealing cannabis in the past. They changed as they became parents, leading to our last theme.

### 3.5. A Paradoxical Desire for Exemplary Parenting

Most of the participants had children and wanted to be good parents. They tried to combine their cannabis use with their role as parents. P1 stressed that she was a good mother and did not smoke around her children.

P1: *“I only smoke in the evening. During the day with my children, it’s forbidden [...] I’m telling you, I only smoke when my daughters are in bed because it allows me to get high and not have to say to myself, ‘Oh, the little ones, what if one gets up?”*

This was not only a maternal reaction but also a paternal one. P3 had stopped using cannabis to support his wife: *“as soon as I knew my partner was pregnant [...] I wanted to have clear ideas of how to accompany her during her pregnancy and [...] to handle my child.”*

Participants were responsible parents and their children became their priority. Parenthood was difficult and they recognized the difficulty of their role.

P9: *“Something very strong happens when you become a mother and [...] Everything is put back in its place, I don’t know how to say it, but the accessory becomes accessory again and is no longer in the foreground. On the other hand, the essential is finally recognized as being the essential.”*

They refused that their children would go on to repeat their mistakes. They wanted to be better at parenting than their parents had been, to be more open but paradoxically resemble their parents whom they admire. The theme of ambivalence was also seen again.

For P6, *“Parenthood replaces a lot of drugs.”*

## 4. Discussion

### 4.1. Main Findings

To our knowledge, this was the first recent study to describe the reasons and views of adults for continuing their cannabis consumption, as medical cannabis, from 30 years of age onwards. Earlier studies, most of them in the early 2000s, did not specifically study medical cannabis, but rather the perception of cannabis in all age groups, specifically among adults [20,21]. In our study, in-depth interviews were conducted on the feelings and emotions of the participants, allowing for them to describe their personal stories and experiences. All of the reasons found for using “medical” cannabis in adulthood were experienced by most of the participants.

The use of cannabis to sooth traumatic experiences has been previously detailed in the literature. Indeed, a systematic review and meta-analysis showed that sexual and physical abuse during childhood were factors affecting vulnerability to cannabis use in adolescence, whilst also associating with substance use in adulthood [22]. As it is impossible to appease certain violent external triggers, cannabis is often used to provoke an internal appeasement, as being a form of resignation beyond a certain resilience.

In these situations, cannabis is used as an antidepressant. Nevertheless, this role of cannabis as an antidepressant is controversial in the literature [23,24,25]. A recent, randomized trial showed that cannabis improved insomnia but had no effect on anxiety or depression [26]. Cannabis is said to have euphoric and anxiolytic effects, but with low symptoms and improvement in the quality of life [27]. Anxious and depressed people and cannabis users do not have the same level of activation of brain areas [27], which could explain the lack of effect of cannabis.

Cannabis is also seen as a natural, organic product, a notion found in the ARAMIS qualitative study [15]. However, our interviews did not only reveal a more or less regular consumption of cannabis, but also the concept of addiction, which may or may not have been present among the interviewees. Addiction is a normal reaction to an abnormal situation. This notion of addiction, far from being recent, was defined as “the repetition of acts likely to provoke pleasure but marked by dependence on a material object or a situation sought and consumed with ‘greed’” [28]. Furthermore, the notion of psychological trauma is found in “the question of considering the emotional deficiencies that lead the addict to pay with his or her body for the unfulfilled commitments contracted elsewhere” [29]. This definition reflects a traumatic reality underpinned by early deficiencies in one’s childhood. The idea of a situation originating in childhood is found, as an addicted person is defined as a slave to a single solution to escape mental pain [30]. This element was observed in some participants in our study.

There are works, recommendations and labelled diagnostic and therapeutic tools to help in the detection of unhappiness in general medical practice [31,32]. It is a way to alleviate addictions and the possible suicidal thoughts or self-harm of the patient, in order to care for teenagers at risk.

Participants’ ambivalent relationships with close family and friends could be conceptualized by the Karpman drama triangle [33] which identifies three roles: the rescuer, the persecutor and the victim. A parallel with cannabis users can be drawn for users who could be considered as victims. In this context, cannabis would be both a savior, that makes one feel better and more relieved, and a persecutor, because it created an addiction one fights against. The same drama triangle could be observed in participants’ relationships with their loved ones. Parents, brothers or sisters and even friends could act as both rescuers, in the role of supporters, and persecutors, because they may judge participants about their cannabis consumption. They sometimes even assumed the role of the tormentor, and participants’ reason for their ill-being (parents’ divorce, criticism of their weight, rejection because of their difference). This dramatic triangle helps to explain the situations of inner conflict in which the participants found themselves.

To get out of this triangle, different approaches have been described. The Empowerment Dynamic (TED), published in 2009 [34], suggests that the victim may adopt the role of “creator” and consider the persecutor as a “challenger”, calling upon a “coach” rather than a rescuer. The “coach” would then help the person to make informed choices. This person could be their wife, such as P6 who said his wife was his “salvation”, the one who enabled him to get over his addiction. Similarly, P9, a psychologist, stopped her use at the same time she started her psychotherapy sessions. Her psychotherapist can thus be seen as her “coach.”

They deal with their use, their entourage’s opinion of their use, but also their role as parents, which is very important for them. Thus, we find, in their discourse, the notion of parenthood oriented around several axes including: the experience of parenthood and the feeling of parenthood clearly explained by the interviewees. They put a certain amount of pressure on themselves to be good parents, which is a contemporary injunction [35,36].

### 4.2. Limitations and Future Research

Our study had some limitations. The first is the small number of subjects. Nevertheless, the size of the sample was sufficient enough for deep analysis and was close to a classic sample in IPA [16]. Second, we interviewed “medical” cannabis users according to a specific definition. The definition of medical cannabis was taken from the literature [35,36,37]. However, as the interviews progressed, answers such as “to forget life’s problems”, “to fit in with a group”, “to do what others do”, “to fill a void” could be interpreted as medical reasons and not only “recreational” cannabis use. So, the line between medical and recreational cannabis is thin and difficult to formally define. Nevertheless, the definition used corresponds unambiguously to medical problems. Only one interviewee, among the 36 “medical” users, consumed cannabis as CBD oil to relieve muscle pain related to multiple sclerosis, instead of smoking joints as all the other participants did. It would be interesting to investigate more participants using CBD alone. It would also be interesting to interview people with a defined “recreational” use of cannabis to compare their reasons with those using cannabis for “medical” reasons. Third, our sample was not completely representative of the general population. Participants were of a high socio-economic level. This is the case for the entire TEMPO cohort. Indeed, populations with a low socio-economic status are generally underrepresented in health studies [38,39]. Moreover, there were no statistical associations between socio-economic status and cannabis use [40] and there is no literature indicating that specific reasons for medical cannabis use may be different according to the socio-economic status. Fourth, the interviews were mainly conducted by video call, which may have hindered the relationship between the participant and the interviewer, but the richness of the interviews suggests this was not a significant limitation.

Our study also has several noteworthy strengths. First, we used a logbook to deconstruct the researchers’ subjectivity. Secondly, we checked that our study design conformed to 30 out of the 32 criteria of the COREQ checklist for reporting qualitative research [17]. The two not-validated criteria concerned the feedback to participants on the transcripts and the use of video calls. The feedback to participants could not be carried out for ethical reasons. We felt that it would be difficult to show participants that their traumatic experiences could have an influence on their use of cannabis. Because of the COVID-19 pandemic, the interviews were mainly conducted via video call, which could be thought to have hindered the relationship between the informants and the interviewer, but the interviews were in-depth enough for analysis. Moreover, a recent article showed that online focus groups could be a good opportunity for studying addictive online behaviors [41]. It might be the same for individual online interviews. Thirdly, the duration of the interviews made it possible to extract very rich information, allowing for better understanding the reasons for the medical use of cannabis by adults. Moreover, the comprehensive interviews helped to limit the desirability bias. Fourthly, our population had a high level of education that allows for them to have a high level of literacy and to take a step back from what medical cannabis is [42].

## 5. Conclusions

Amongst participants, there were various reasons for the continued use of cannabis into adulthood, with the need for internal appeasement from cannabis stemming from difficulties in processing previous violent, external events. Knowledge of the reasons for the consumption of cannabis in adulthood allows for us to better target prevention campaigns. Moreover, it seems important to raise awareness among caregivers so that they may know how to recognize the suffering of adolescents in order to prevent subsequent psychoactive substance use.

## Figures and Tables

**Table 1 ijerph-20-04462-t001:** Characteristics of the study participants.

Participants	Sex	Age	Diploma	Socio-Professional Category	Marital Status	Employment Status	Place of Residence	Residential Area	Chronic Diseases	Forms of Consumption	Type of Consumption
1	Female	34	General Certificate of Secondary Education	Jewelry sales advisor	Married	Unemployed	House	Urban	Crohn’s disease and DT2	General Certificate of Secondary Education	Recent regular use
2	Female	40	Bachelor’s Degree	Employee in a social support service	Married	Unemployed	Family house	Urban	No	Joint and CBD	Recent regular use
3	Male	41	Master’s Degree	Employee in a digital agency	Married	Active	Apartment	Urban	No	Joint	Former consumption
4	Female	39	Master’s Degree	Executive in the pharmaceutical industry	Single	Active	Apartment	Urban	No	Joint	Recent occasional use
5	Male	42	Master’s Degree	Information systems consultant	Married	Off work	Apartment	Urban	Acromegaly	Joint	Former consumption
6	Male	44	Bachelor’s Degree	Railway worker-runs a maintenance workshop	Married	Active	House	Urban	No	Joint	Former consumption
7	Female	43	Bachelor’s Degree	Nurse	Married	Active	House	Urban	Multiple sclerosis (ALD)	CBD	Recent regular use
8	Female	43	Bachelor’s Degree	Executive in a technical cooperation agency	Married	Active	Apartment	Urban	No	Joint	Recent occasional use
9	Female	44	Master’s Degree	Psychologist	Married	Active	Apartment	Rural	No	Joint	Former consumption
10	Male	38	Master’s Degree	Computer scientist	Married	Active	House	Urban	No	Joint	Former consumption
11	Male	44	Bachelor’s Degree	IT Consultant	Married	Active	House	Rural	No	Joint	Recent regular use
12	Female	41	Bachelor’s Degree	Bank executive	Married	Active	House	Urban	No	Joint	Recent occasional use

**Table 2 ijerph-20-04462-t002:** Superordinate themes and categories.

Superordinate Themes	Categories
Soothing a traumatic experience through cannabis use	Unhappiness during adolescence
	Traumatic events
	Cannabis as an antidepressant
An ambivalent relationship with the user and cannabis and with the user and close relatives	An ambivalent relationship with cannabis
	A “false” appeasement?
	An ambivalent relationship with closest relatives
Cannabis, a known soft drug comparable to alcohol or tobacco, leading to an illogical demonization	Being knowledgeable
	Cannabis, a soft drug
	A demonization deemed illogical
Recreational use in the context of experimentation	Recreational use
	Experimenting with other drugs
A paradoxical desire for exemplary parenting	Being good parents
	An essential role, difficult to assume

## Data Availability

These data are not publicly available due to ethical restrictions and the need to preserve participants’ privacy. The datasets used and/or analyzed during the current study are available from the corresponding author on reasonable request.

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
