# Peer review of "Experience of Cannabis Use from Adolescence to Adulthood in France: An Interpretative Phenomenological Analysis"

_ijerph, 2023, doi:10.3390/ijerph20054462_

Round 1

Reviewer 1 Report

1- In the present manuscript, the authors have addressed the use of medical Cannabis in the adult population.

2-The current research is relevant to cannabis research because medical cannabis might be addictive in long-term use and should be addressed.

3- This manuscript specifically addresses the question of prolonged medical cannabis use and previous studies have addressed cannabis use and addiction in general.

4- This Study provides an important finding of long-term medical cannabis use and dependency on medical cannabis.

5- Although, the authors performed the analysis in a small group of patients, the analysis of the study was done in great detail and will be beneficial to the readers and for further studies.

I recommend the acceptance of the manuscript with minor spelling and language checks

Reviewer 2 Report

The current study aims to describe the reasons and motivations for using medical cannabis among adults over 30 years old.

The authors performed an interpretative phenomenological analysis and people with a history of or current use of cannabis were recruited from the TEMPO cohort.

The authors should be congratulated for their work and for addressing an important topic. Only a few points warrant mentions:

Major comments:

1.             In the “Conclusion” section, it would be fair if authors could report conclusions related to their aim: reasons and motivation for using medical cannabis in adults over 30 years old. The last sentence should be replaced in this setting; indeed, the last sentence of the conclusions would be more accurate in the “Discussion” section as part of future works.

Minor comments:

1.             In the “Discussion” section, as the study deals with topics such as diseases of recruited participants, parenting, “illogical” demonization of cannabis. It would be interesting if authors could mention recent clinical findings of the effects of the recreational use of cannabis on general health, such as on the fertility potential of men as in PMID: 35868833

Reviewer 3 Report

The title gives an impression of a broad study of a current trend in marijuana use. The abstract also gives a general description but does state the intended approach of an interpretive phenomenological analysis on 12 participants. The study however is much narrower with participants coming from France and are educated and fully employed backgrounds. The study states that France has more restrictive measures on marijuana but does not provide these nor make a strong comparison to other countries in the EU that are not as restrictive. The focus on the dozen participants does not really provide a strong support to the claims made by the researchers. The reasons are not compelling enough to read about why these adults are using marijuana. The interpretive part of the research does not come through. There needs to be a deeper explanation and analysis. This may occur through reference to other research and work that show why adults us illicit drugs. Or, this could be compared to addictive behaviors. This reader is not sure of the five superordinate points highlighted as the reasons for the use of cannabis. 

An explanation of an interpretive phenomenological analysis is needed. Taking quotes from the interviewees and making any further observation is not an analysis. The opportunity to give an interpretation or even a stronger analysis is missing from this paper. There doesn't seem to be any compelling reason for a reader to want to read this after reading the abstract. It is just too broad and doesn't really give a strong thesis. 

Round 2

Reviewer 3 Report

The article is greatly improved from the first draft. The suggestions and comments made were taken into consideration. The article has potential to be expanded on and go more in depth.